# *Nemesignis*, a Replacement Name for *Nemesis* Furfaro & Mariottini, 2021 (Mollusca, Gastropoda, Myrrhinidae), Preoccupied by *Nemesis* Risso, 1826 (Crustacea, Copepoda)

**DOI:** 10.3390/life11080809

**Published:** 2021-08-10

**Authors:** Giulia Furfaro, Paolo Mariottini

**Affiliations:** 1Department of Biological and Environmental Sciences and Technologies—DiSTeBA, University of Salento, I-73100 Lecce, Italy; 2Department of Science, University of Roma Tre, I-00146 Rome, Italy; paolo.mariottini@uniroma3.it

**Keywords:** ICZN, *Dondice*, homonym, Heterobranchia

## Abstract

The genus *Nemesis* Furfaro & Mariottini, 2021, was recently introduced for an independent lineage of aeolid nudibranchs, and *Dondice banyulensis* Portmann & Sandmeier, 1960, established as its type species. Anyway, the presence of a senior homonym, *Nemesis* Risso, 1826, was evidently missed. In fact, in 1826, Risso established this genus for a group of Copepoda (Arthropoda, Crustacea) and according to the Principle of Priority (ICZN) only the senior homonym may be used as a valid name. Therefore, a new replacement name is here proposed. Furthermore, the genus name *Nanuca* Er. Marcus, 1957, has priority over *Dondice* Er. Marcus, 1958 and consequently, the species in this clade should be classified under *Nanuca*, mostly as new combinations.

## 1. Introduction

The genus *Nemesis* Furfaro & Mariottini, 2021, was introduced based on evidence from a recent integrative systematic study [1] (Furfaro & Mariottini, 2021) for an independent lineage of aeolid nudibranchs, and *Dondice banyulensis* Portmann & Sandmeier, 1960 [2], was established as the type species. The newly identified lineage is currently monospecific and characterized by (i) the central cusp of the radular tooth that is not marked and a little longer than lateral denticles, (ii) the long distal and proximal deferent ducts of the male portion of the reproductive system and (iii) its inability to autotomise the cerata when stressed by possible predators. Just after the publication of our paper, Luigi Romani (Lucca, Italy) sent to us a letter (e-mail: 18.06.2021) where he noted that in our recent manuscript, we have evidently missed the existence of a senior homonym, *Nemesis* Risso, 1826 [3] (International Commission on Zoological Nomenclature - ICZN, 1999: Article 53.2) [4]. In fact, in 1826, Risso established this genus for a group of Copepoda (Arthropoda, Crustacea) and according to the Principle of Priority (ICZN, 1999: Article 52.3) [4], when two or more names are homonyms, only the senior may be used as a valid name. Therefore, a new replacement name is here proposed under the Article 60.3 of ICZN. Furthermore, Philippe Bouchet (Paris, France) pointed out to us that *Nanuca* Er. Marcus, 1957 [5], has priority over *Dondice* Er. Marcus, 1958 [6]. Consequently, the species in this clade should be classified under *Nanuca*, mostly as new combinations.

## 2. Results and Discussion


**Taxonomy**


Familia Myrrhinidae Bergh, 1905 [7]

Genus *Nemesignis*
**nom. nov.** pro *Nemesis* Furfaro & Mariottini 2021 (non Risso, 1826)

urn:lsid:zoobank.org:pub:DEED49D6-F89B-4D68-A8A3-E1071197264C

**Type species.***Dondice banyulensis* Portmann & Sandmeier, 1960.

**Etymology.** The genus name *Nemesignis* comes from the union of the Greek word *Nemesis*, that recalls the homonymous Greek goddess and her role of compensatory justice, with the Latin word *Ignis*, that is the fire that burns and blazes, linked to the fiery red colour of the type species of the genus.

**Included species.***N. banyulensis* (Portmann & Sandmeier, 1960).

Genus *Nanuca* Er. Marcus, 1957

**Type species.***Nanuca sebastiani* Er. Marcus, 1957

= *Dondice* Er. Marcus, 1958 (type species: *Caloria occidentalis* Engel, 1925 [8])

**Included species.***Nanuca sebastiani* Er. Marcus, 1957, *Nanuca galaxiana* (Millen & Hermosillo, 2012) **comb. nov.** [9], *N. occidentalis* (Engel, 1925) **comb. nov.**, *N. parguerensis* (Brandon & Cutress, 1985) **comb. nov.** [10], *N. trainitoi* (Furfaro & Mariottini, 2020) **comb. nov.** [11].

## 3. Conclusions

The presence of a senior homonym, which has priority over the recently stated *Nemesis* Furfaro & Mariottini, 2021, made this latter genus name as invalid and invoked the need for a replacement name according to the rules of the ICZN. Therefore, *Nemesignis*
**nom. nov.** is here proposed as the new replacement name, under the Article 60.3 of ICZN and consequently, *Nemesignis banyulensis* (Portmann & Sandmeier, 1960) is its type species. Finally, since *Nanuca* Er. Marcus, 1957 has priority over *Dondice* Er. Marcus, 1958, the species in this clade should be classified under *Nanuca*, as *Nanuca galaxiana* (Millen & Hermosillo, 2012) **comb. nov.**, *N. occidentalis* (Engel, 1925) **comb. nov.**, *N. parguerensis* (Brandon & Cutress, 1985) **comb. nov.**, *N. trainitoi* (Furfaro & Mariottini, 2020) **comb. nov**.

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
