# Peer review of "Nemesignis, a Replacement Name for Nemesis Furfaro & Mariottini, 2021 (Mollusca, Gastropoda, Myrrhinidae), Preoccupied by Nemesis Risso, 1826 (Crustacea, Copepoda)"

_life, 2021, doi:10.3390/life11080809_

Round 1

Reviewer 1 Report

The genus Nemesis Furfaro & Mariottini 2021 is a junior synonym for Copepoda's genus Nemesis and should be changed by ICZN's Principle of Priority (ICZN, 1999: Article 52.3). So the authors refer to genus Nemesis Furfaro & Mariottini 2021 as Nemesignis nom. nov. It was well structured with essential content in naming, and no major criticism was found in the manuscript. The content of the manuscript is appropriate and well written. There is nothing to edit, so I suggest accepting this manuscript as it is.

Author Response

Thank you for your comment and for your help.

Best regards.

Reviewer 2 Report

Review of Nemesignis, a replacement name for Nemesis Furfaro & Mariottini, 2021 (Mollusca, Gasteropoda, Myrrhinidae), preoccupied by Nemesis Risso, 1826 (Crustacea, Copepoda) by Furfaro and Mariottini for Life

I will recommend the short note for acceptance pending the correction of a few typos and copy-editing as needed, as well as the adding of some missing citations.

Here are some typos I’ve noticed:

Line 11 – “evidences” should be “evidence”

Line 13 – remove the comma before “and” and add “was” before “established”

Line 33 – the first letters of “compensatory” and “justice” should be lower case

The following citations should be added to the “References” section:

Risso, 1826

Portmann & Sandmeier, 1960

Marcus, 1957

Marcus, 1958

Bergh, 1905

Millen & Hermosillo, 2012

Engel, 1925

Brandon & Cutress, 1985

Furfuro & Mariottini, 2020

It is important to include these taxonomic citations in this situation in order to contextualize the oversight that brought about the publication of this short note.

Author Response

We have corrected the text according to all your kind suggestions and we have added all the references you have mentioned.

Thank you for your very kind revision that, for sure, helped us to improve the quality of our short communication.